# Edible Halophytes and Halo-Tolerant Species in Apulia Region (Southeastern Italy): Biogeography, Traditional Food Use and Potential Sustainable Crops

**DOI:** 10.3390/plants12030549

**Published:** 2023-01-25

**Authors:** Rita Accogli, Valeria Tomaselli, Paolo Direnzo, Enrico Vito Perrino, Giuseppe Albanese, Marcella Urbano, Gaetano Laghetti

**Affiliations:** 1Department of Biological and Environmental Sciences and Technologies (DiSTeBA), Salento University, 73100 Lecce, Italy; 2Department of Biosciences, Biotechnologies and Environment, University of Bari “Aldo Moro”, 70126 Bari, Italy; 3Institute of Biosciences and BioResources (IBBR), National Research Council (CNR), 70126 Bari, Italy; 4CIHEAM, Mediterranean Agronomic Institute of Bari, Via Ceglie 9, 70010 Valenzano, Italy

**Keywords:** halophytes, edible wild species, coastal areas, *Salicornia*, *Arthrocaulon macrostachyum*, *Soda inermis*, *Cakile maritima*, *Crithmum maritimum*, *Reichardia picroides*, *Silene vulgaris* subsp. *tenoreana*, *Allium commutatum*, *Beta vulgaris* subsp. *maritima*, *Capparis spinosa*

## Abstract

The Mediterranean basin is rich in wild edible species which have been used for food and medicinal purposes by humans throughout the centuries. Many of these species can be found near coastal areas and usually grow under saline conditions, while others can adapt in various harsh conditions including high salinity. Many of these species have a long history of gathering from the wild as a source of food. The aim of this contribution is an overview on the most important halophyte species (*Salicornia* sp. pl., *Arthrocaulon macrostachyum* (Moric.) Piirainen & G. Kadereit, *Soda inermis* Fourr., *Cakile maritima* Scop., *Crithmum maritimum* L., *Reichardia picroides* (L.) Roth., *Silene vulgaris* (Moench) Garcke subsp. *tenoreana* (Colla) Soldano & F. Conti, *Allium commutatum* Guss., *Beta vulgaris* L. subsp. *maritima* (L.) Arcang., *Capparis spinosa* L.) that traditionally have been gathered by rural communities in southern Italy, with special interest on their ecology and distribution, traditional uses, medicinal properties, marketing and early attempts of cultivation. It is worth noting that these species have an attractive new cash crop for marsh marginal lands.

## 1. Introduction

The relationship between human civilizations and salinity has existed for thousands of years. The total area of saline and sodium lands is likely to be approximately 10% of arable land worldwide [1]. Many of the factors that lead to soil salinization are being exacerbated by climate change and it will get worse and worse over the next few years based on the indicators that the scientific world takes into consideration. In fact, considering the increase of global climate change and severe conditions prevailing all over the world, conventional crop cultivation is facing various limitations related to shortage of good quality water, rising temperature and the salinization and degradation of soil properties, especially in arid and semi-arid regions of the Mediterranean basin where the aforementioned problems intensify [2]. Therefore, it is of the utmost importance to propose alternative crop species able to adapt to difficult conditions in the framework of saline agriculture and constitute good candidates as potential food and medicinal crops [3], as already recently enacted in Apulia for some wild aromatic coastal species of *Lamiaceae* family [4,5,6] that can meet the future needs of populations who will face this challenge.

Halophytes are defined as plants capable of developing and completing their biological cycle in natural saline environments with concentrations greater than 200 mM NaCl [7]. They constitute a highly specialized flora adapted to hypersaline environments. The response to this type of stress is manifested through osmotic adjustments, morpho-anatomical and physiological adaptations. Most of these species are eu-halophytes, which are obligate halophytes because they grow and develop in media with high salt concentrations. Often, they are succulent plants, since they have large amounts of water in their tissues in addition to accumulating salts and ions, maintaining the osmotic balance [8]. In recent years, the nutritional interest of these genera has sparked an increase in research articles on elemental composition, minerals, bioactive compounds, polyphenols, fatty acids and flavonoids [9,10]. Halophytic species could be used for the saline agriculture proposed by [11]. The types of land proposed for use in halo-culture are generally internal and coastal marginal uncultivated lands, degraded saline lands, sandbanks and salinized lands in general which are not able to economically produce conventional agronomic crops. 

There are two factors that make halophytes of special interest to be considered in the food industry: first, their economic potential, since their productivity in high-salinity and low-water intake environments is much higher than that of traditional edible species; and second, their nutritional value in terms of their protein, phenolic, lipid contents and the great quantity of minerals such as potassium, calcium, and magnesium and other bioactive compounds [12,13,14]. Several halophyte species are already being used as food, forage, oilseeds and medicines [15,16,17,18]. 

Examples of edible halophytes and halo-tolerant species include marine fennel (*Crithmum maritimum* L.) [19], Mediterranean saltwort (*Soda inermis* Fourr.) [20], glassworts (*Salicornia* sp. pl.; *Arthrocaulon macrostachyum* (Moric.) Piirainen & G. Kadereit) [21,22], sea rocket (*Cakile maritima* Scop.), sea beet (*Beta vulgaris* L. subsp. *maritima* (L.) Arcang.) [23], *Reichardia picroides* (L.) Roth. [24], caper (*Capparis spinosa* L.) [25], *Silene vulgaris* (Moench) Garcke subsp. *tenoreana* (Colla) Soldano & F. Conti and maritime wild leek (*Allium commutatum* Guss.) [26]. 

This review describes the edible halophytes and halo-tolerant species above reported in respects of their ecology and distribution, traditional uses, medicinal properties, marketing and early attempts of cultivation. Moreover, original new data are provided about the distribution of their use in the Apulia region (southeastern Italy) and about some early attempts of domestication/cultivation).

## 2. Materials and Methods

### 2.1. Study Area

The Apulia Region, located in the southeastern part of the Italian peninsula, has a surface area of more than 19,000 km², and is geologically characterized by Cretaceous limestones and calcarenites and by alluvial deposits (Pliocene–Pleistocene). It is the Italian region with the highest coastal extent (Figure 1), with about 1224 km of coastline (including the Tremiti Islands) stretched between the Adriatic and the Ionian Seas, from RodiGarganico to Ginosa Marina [27]. Throughout this paper we refer mostly to Gargano (northern Apulia) and Salento (southern Apulia) areas. This particular geographic conformation has induced uses and traditions in the Apulian people, who express a strong relationship of knowledge and uses of the coastal area not only for commercial exchanges but also for craft activities such as the extraction of building materials, the tanning of leather, wool washing and fabric dyeing [28] and, most importantly, the extraction of salt from sea water [29]. In relation to the latter, the most important extraction sites are (or have been) Salina di Margherita di Savoia (FG), Salina di Punta della Contessa (BR) and Salina dei Monaci di Manduria (TA); however, almost all the low Apulian coasts have established local resources for the supply of salt. The halophytes, growing in these coastal environments, have been an important asset for maritime populations, as source of mineral salts and numerous secondary metabolites precious for the human diet [30,31].

The largest part of the region (over 80%) is used for agriculture and, as in other arid and semi-arid Mediterranean areas, has continuously been subjected to intensive management practices, potentially leading to land degradation [32,33]. As far as the plant biodiversity is concerned, Apulia is an area of high biogeographical interest and high floristic density [34,35], with a fairly large number of endemic taxa [36]. Similar diversity and richness are found in terms of plant communities, with a vegetational variability emphasized by a long-time exploitation of the area that has produced, in time, numerous, alternating and often contrasting landscapes. In particular, coastal areas, for their wide extent and geomorphological variability, are characterized by a high floristic richness [37] and variety of plant landscapes. Many phytosociological surveys have been carried out along the Apulian coast, on both rocky and sandy shores included salt marshes [38,39,40,41,42,43,44,45,46,47,48,49,50,51,52]. According to the most recent literature, the vegetation of the rocky coasts is classified in several associations of the orders *Crithmo-Staticetalia*, *Helichrysetalia italici* and *Senecionetalia cinerariae* (*Crithmo-Staticetea* class) [50] for perennial communities, and of *Saginetea maritimae* (*Saginetalia maritimae*) for annuals; salt marsh vegetation is framed, with numerous associations and alliances, within the classes *Thero-Salicornietea* (*Thero-Salicornietalia* and *Thero-Suaedetalia splendentis*), *Saginetea maritimae* (*Frankenietalia pulverulentae*), for the annual vegetation [51] and, for perennial vegetation, within the *Salicornietea fruticosae* and *Juncetalia maritimae* [45,46,47,48]. Finally, the vegetation of sandy coasts falls within the *Cakiletea maritimae* and the *Ammophiletea* classes [43,53]. These plant communities are included as habitats of community interest within 92/43 EEC “Habitat” Directive [54,55,56,57,58,59].

### 2.2. Bibliographic Review; New Data Collection in the Field

More than 50 bibliographic sources published in the last 40 years by Apulian publishers and referring to different uses and properties of local plants were retrieved and examined [60,61,62,63,64,65,66,67,68]. Information was derived from numerous surveys carried out in the last twenty years in various territories of the Apulia region, by interviews with the elderly people and with “connoisseurs” or “experts” of wild herbs. Similarly, in the last year, specific explorations in the most significant coastal areas of the region were carried out, to evaluate the presence of halophilic species of ethnobotanical interest. For each site, residents were contacted and site inspections were carried out in the coastal places they usually visit for herb collections, with the identification of the species known to them and recording, on semi-structured questionnaires, of information about: (a) “plant” (scientific and common name, habitat, collection period and site, used part, etc.); (b) “use” (e.g., human nutrition, animal food, folk medicine, etc.); and (c) “preparation, proverbs, recipes”. The questionnaire structure is showed in Appendix B. Interviews were also addressed to restaurant managers in the most popular coastal locations and to fishermen in the most traditional fishing grounds. About eighty-five questionnaires were administered, and the collected data were then organized in a database. Moreover, extensive bibliographic research on food use and domestication/cultivation of the considered species also in other geographical areas, as well as other uses, has been carried out and reported in Results. 

For each of the investigated species, we reported: brief description, distribution and ecology, food use, domestication/cultivation and other uses/properties.

As regards taxonomical nomenclature of the treated species, we followed the Italian checklist [69]. For the genus *Salicornia* (annual species), we followed “Flora d’Italia” [70]. For the syntaxonomical nomenclature of vegetation types mentioned in “distribution and ecology”, we referred to the specialized literature reported in the previous section, “Study area”.

### 2.3. Taxonomical Notes

*Salicornioideae* (Amaranthaceae/Chenopodiaceae *sensu* APG IV [71]; subfamily for succulent, articulated and apparently leafless species) stands out by comprising exclusively succulent hygro-halophytes with highly specialized morphological, anatomical and physiological adaptations to their environment in coastal and inland halophytic communities [72,73,74]. Within the *Salicornioideae*, the genus *Salicornia* (glassworts) displays a controversial taxonomical classification; this group has been thoroughly analyzed from both morphological and molecular traits. Common *Salicornioideae* characters are their succulent, articulated and apparently leafless stems, and the spike-like inflorescence of sessile, 3-flowered cymes, reduced flowers, usually consisting of a 2–4 lobed calyx tube with 1–2 stamens, and the sub-annular or curved embryo [75,76]. The genus *Salicornia* was for a long time considered as circumscribed to annual species, with the perennial species separated in other genera, but recent taxonomical revisions [74] provided a new framework, with the (perennial) *Sarcocornia* species included under *Salicornia* genus; thus, the *Salicornia* genus currently frames both annual and perennial species. In the same contribution, *Arthrocnemum macrostachyum* (Moric.) K. Koch is treated in a different genus (i.e., *Arthrocaulon*) as *Arthrocaulon macrostachyum* (Moric.) Piirainen & G. Kadereit.

In general, the taxonomic identity of the *Salicornia* species is a challenging issue. The complexity of the genus *Salicornia* in Europe has produced many reversals, alternating partitions within the genus. As regards the annual *Salicornia* species, two series have been traditionally identified, diploid and tetraploid, each of them with numerous species and micro-species. In the revision of [75], only three entities are recognized in the Mediterranean: *Salicornia procumbens* subsp. *procumbens* (tetraploid, distributed along Mediterranean and Atlantic coasts and including *Salicornia emerici*, *Salicornia veneta* and *Salicornia dolichostachya*), *Salicornia perennans* subsp. *perennans* (diploid, with Mediterranean and Eurasian distribution, including *Salicornia patula*) and *Salicornia europaea* subsp. *europaea* (diploid, western Mediterranean). As regards the Italian peninsula, only the two first taxa, *S. procumbens* subsp. *Procumbens* and *S. perennans* subsp. *perennans*, are recognized. Nevertheless, the existence of numerous microtaxa that are morphologically quite well-differentiated and characterized by particular ecological conditions would encourage them to maintain their use. For this reason, we refer to the classification proposed by [70], who recognizes the presence in the Italian peninsula (and the Apulia region) of the diploid *S. patula* and of the tetraploid *S. emerici*, *S. veneta* and *S. dolichostachya*.

Species of this group are characterized by the plasticity of their phenotype which, along with the reduction of their leaves and flowers and the difficulty of preserving the dry specimens in the herbaria, complicates the separation of the species with frequent confusion among *Sarcocornia* taxa [76]. The taxonomic complexity of this group implied that, in the course of our investigations, the identification of the correct taxonomic entity has often been problematic, especially when two or more related species coexist in the same site (e.g., *S. emerici* and *S. dolichostachya* or *Salicornia fruticosa* and *Salicornia perennis*).

## 3. Results

### 3.1. Salicornia sp. pl. (Annual)

#### 3.1.1. Brief Description, Distribution and Ecology

*Salicornia*, also commonly known as pickleweed, glasswort, sea asparagus and samphire, derives its name from the Latin word meaning “salt” and includes strictly halophytes. These taxa accumulate inorganic salts and water in their stems [8]. The most common elements found are Na, Ca, K and Mg, among others, and are present in the stem and roots [9]. These species grow usually at the edges of wetlands, salt marshes and mudflats world-wide (except for Australia). Along with the perennial *Salicornia* sp.pl. and other perennial succulent Amaranthaceae/Chenopodiaceae, they thrive in littoral and coastal salt marsh or in inland salt pans, forming distinct vegetation types subjected to different flooding regimes with brackish-to-saline water (depending on tidal influences) and usually composed of almost monospecific plant communities. In general, annual *Salicornia* species occur in the innermost parts of the salt marshes, subject to longer periods of submersion. This group of species is characterized by succulent and articulated stems with opposite pairwise-fused leaves and bracts and inconspicuous flowers with three-flowered cymes in which the lateral flowers are in contact below the central flower. The green plant turns orange, pink to reddish in autumn, before dying in winter [75,76,77]. In the Apulia region, four species have been recognized: the diploid *S. patula* Duval-Jouve (Figure 2a), and the tetraploid *S. emerici* Duval-Jouve, *S. veneta* Pignatti & Lausi and *S. dolichostachya* Moss (Figure 3a) [70], often very similar morphologically to each other and distinguishable with certainty only if observed in full maturity (flowering and fruiting period). These species typically grow forming the pioneer coastal or continental vegetation of maritime and inland salt marshes, occupying those sites with the highest salt concentration, generally positioned in the first-line belt of the flooding zonation (Figure 2b and Figure 3b) on raw soils from sandy to loamy or clay, depending on sedimentation conditions and in large part are poor in nutrients, temporarily flooded and drying out in summer (*Thero-Salicornietea* Tx. in Tx. & Oberd. 1958; *Salicornion patulae* Géhu & Géhu-Franck 1984; *Salicornion venetae* Tomaselli et al. 2020). The phenological optimum is late summer to autumn [51]. According to the 92/43/EEC “Habitat” Directive, this vegetation falls within the habitat of community interest 1310—“*Salicornia* and other annuals colonizing mud and sand”.

#### 3.1.2. Food Use

The annual glassworts are almost all of food interest. Like other chenopods such as *Sarcocornia perennis*, *S. fruticosa* and *Arthrocaulon macrostachyum* recently introduced for human consumption [78], *S. patula* and other annuals are also an excellent candidate due to the mineral content. Actually, they are little known in the territories of southern Salento where, due to the rocky nature of the coasts, temporary salt ponds and salt marshes have sporadic presence and very limited extent. On the contrary, annual glassworts are widely used in the areas characterized by coastal salt marshes (e.g., Gargano, Capitanata, Brindisi and Taranto areas). The plants are low and herbaceous in consistency, and the tender tips of the young branches are easily broken by hand. The most abundant harvest is in spring, before the reproductive stage begins. The collected tips are cleaned, washed and boiled. After having drained them, one by one, the fleshy green part is removed (like a glove) and the central woody part is thrown away. They can be dressed with oil and lemon or vinegar, put in jars in oil or vinegar or frozen and used as needed. In the area of “Mar Piccolo”, near Taranto, in summer it is used to mix the sea asparagus with leavened dough, which is then portioned and fried (“pettole”). Recently, in the Gargano area it is common to find pizzas or pasta topped with sea asparagus (Figure 4 and Figure 5c; Table 1; Appendix A). In recent years, the sale of sea asparagus in oil or in vinegar has spread in various locations in northern Apulia (especially Gargano) and in the Bari area, even in large-scale distribution.

In the Iberian Peninsula and France, *S. emerici* has long been largely consumed as food [79]. In South Korea, tender *Salicornia* shoots are processed in drinks such as nuruk (a type of fermentation initiator), makgeolli (a Korean rice wine) or vinegar [80]; the aerial parts are used in salads, and their consumption is a source of salt in the diet [77].

#### 3.1.3. Domestication/Cultivation

Being therophytes, the only method of propagation is by the generative way (from seeds). The seeds are located in a bract cavity, unprotected, so they fall as they reach maturity. It is necessary to be timely in collecting the parts that have already matured the seeds and to be careful not to drop them during the cutting and bagging operations. The quantity of seeds collected per plant is always minimal. Recently, interest towards *Salicornia* sp. has risen dramatically, especially in the Gargano area where some attempts of medium-scale cultivations have been run [22]).

In Israel, *Salicornia* and *Sarcocornia* cultivation for vegetable production is typically practiced under simple nets or in greenhouses on land areas of 0.5–1 ha. For the products of these crops to be successfully marketed as vegetables, the young shoots must be harvested manually, an intensive element critical to halophyte crop production. Different cultivation protocols, therefore, have been tested for *Salicornia* vegetable production. For example, the easiest and most straightforward way to produce *Salicornia* is to cultivate it in native sand dune soils watered with drip irrigation, as has been successfully accomplished by some farmers in the Dead Sea and Ramat HaNegev areas [81,82].

*Salicornia* sp. pl. is cultivated on a commercial scale in Israel and in Mexico, the United Arab Emirates, Saudi Arabia and India. In many parts of the Europe, the use of this halophyte is mainly based on the harvesting of wild plants, while only in the Netherlands, Portugal and France are the plants of *Salicornia* sp. pl. cultivated. In Italy, the first attempts of *S. patula* cultivation for family consumption started over 40 years ago in the gardens located around the Lesina lagoon (Apulia, Southern Italy) [22]. After only a few years, some local farmers started the cultivation of this wild grass as a cash crop, due to the increasing interest in this food, the attractive price and the possibility of economic profits from the cultivation of marginal lands [22].

In Apulia, harvesting in open fields takes place in July–August. In Israel, it is from August–September, and it is carried out manually in order to maintain the high quality of the final product; only fresh and tender parts can be sold (Figure 5a,b). *Salicornia* can be collected several times during the year in greenhouses. The harvesting can be repeated every two or three weeks (depending on the level of development). Plants are cut above 5 cm from the ground at a height of 10–15 cm. This repeated harvesting enables the same plant to be cut from three-to-four times, depending on the level of growth. The yield can reach 10–15 tons per hectare [9].

#### 3.1.4. Other Uses/Properties

Like other halophilic species, even in glassworts the stress of the alkaline environment induces the synthesis of metabolites that can be useful to humans for their antioxidant and anti-inflammatory power [83]. For example, bioactive compounds such as phenols and fatty acids have been isolated for *S. patula*, and it has been found that the concentration of these compounds depends greatly on how close the plants are to the sea. Caffeic, coumaric, salicylic and trans-cinnamic acids and flavonoids, such as quercetin-3-O-rutinoside, kaempferol/luteolin, apigenin 7-glucoside and pelargonidin-3-O-rutinoside, have been isolated. Furthermore, high concentrations of palmitic, oleic and linoleic acids have been found [84].

### 3.2. Salicornia sp. pl. (Perennial) and Arthrocaulon macrostachyum (Moric.) Piirainen & G. Kadereit

#### 3.2.1. Brief Description, Distribution and Ecology

This group includes species, commonly named as perennial glasswort or perennial marsh samphire, with erect-to-prostrate woody stems, sometimes creeping and rooting at the nodes; that are succulent, articulate and green but becoming reddish to maturity; that have opposite vestigial and long connate leaves clothing the internode, a reduction of vegetative and floral parts and 3–12 flowers hidden in cavities in the inflorescence axis. At present, the following taxa can be recognized in southeastern Italy: *S. perennis* Mill. subsp. *perennis*, characterized by a prostrate and decumbent habit with radicant branches, and a seed testa with scarce, patent, erect, long, fine and occasionally hooked hairs; *S. perennis* Mill. subsp. *alpini* (Lag.) Castrov. with prostrate-to-erect habit and dense, appressed, hooked, occasionally forked, long fine hairs that cover the entire seed testa (Figure 6c,d) and *S. fruticosa*, a 50–150 cm-tall woody shrub with an erect, branchy and pubescent seed exotesta covered mainly on the edges (Figure 6a,b) [76]. *Arthrocaulon macrostachyum* is a woody shrub usually erect, up to 150 cm tall, sometimes prostrate, usually richly branched, with fleshy segments that are glaucous or yellowish green (Figure 7a). 

As for the annual *Salicornia* species, *S. perennis* subsp. *perennis*, *S. perennis* subsp. *alpini*, *S. fruticosa* and *A. macrostachyum* usually grow in coastal salt marsh-forming halophilous shrub plant communities developing on halomorphic soils and spatially arranged in distinct vegetation types (belts) depending on the different flooding period and the water salinity (*Salicornietea fruticosae* Br.-Bl. et Tx. ex A. Bolòs y Vayreda 1950; *Salicornion fruticosae* Br.-Bl. 1933; *Arthrocnemion glauci* Rivas-Mart. et Costa M. 1984; *Suaedion brevifoliae* Br.-Bl. et O. de Bolòs 1958). According to the 92/43/EEC “Habitat” Directive, this vegetation falls within the habitat of community interest 1420—“Mediterranean and thermo-Atlantic halophilous scrubs (*Sarcocornetiea fruticosae*)”. *A. macrostachyum* also grows extensively on rocky coasts forming, with *Crithmum maritimum* and other few species, the first belt of vegetation which is the most exposed to sea spray and waves (Figure 7b,c) [43,47,50,85].

#### 3.2.2. Food Use

The bibliographic evidence on the use of perennial glassworts for food purposes dates back to the 18th century, but these halophytes have been collected and used in Apulia since the Middle Ages [29]. The young shoots are harvested by hand, in autumn or early spring, when they are still tender, breaking them at the point where they begin to lignify. As for the annual glassworts, the shoots are cleaned, washed and boiled; after having drained them, one by one, the central woody part is removed from the green fleshy part, which comes off easily. They can be dressed with oil and lemon or vinegar, put in jars in oil or vinegar or frozen; they are also used in dishes with soups and omelettes or in combination with fish and shellfish as a side dish.

*S. perennis*, *S. fruticosa* and *A. macrostachyum* are also widely consumed in other Mediterranean countries [78,79].

#### 3.2.3. Domestication/Cultivation

In *A. macrostachyum* and perennial *Salicornia* species the seeds are placed in a bract cavity, unprotected, and the same harvesting and sowing techniques used for the annual *Salicornia* species must be used. However, since these are perennial species, with prostrate-ascending stems, it is also possible to adopt agamic propagation, using the branches at the base of the plant, which, in contact with the substrate, already tend to take root under the mother plant. *S. perennis* easily radicates at the nodes, and therefore it is possible to cut the stem before and after the node that generates the plant, eradicate the plant trying to extract the root with a clod of substrate and then transplant it into a nursery pot until it grows. It has been experimented that many halophilic species produce heteromorphic seeds not only in shape, color and size but also for physiological properties. Therefore, different seeds respond differently to environmental conditions, and this is an adaptive strategy that allows the species to adapt to fluctuating conditions. Furthermore, it has been ascertained that in many other halophytes there is no dormancy and that the different types of seeds respond with different germination percentages at the same salt concentration [86].

*S. perennis* is an economically valuable vegetable crop, which can be cultivated in soils where other vegetable plants cannot grow due to high salinity [87]. Some information about the cultivation of perennial *Salicornia* species and *A. macrostachyum* have been reported in the previous section.

#### 3.2.4. Other Uses/Properties

The beneficial effects of *Salicornia* sp. pl. were already known throughout the Mediterranean Basin; for centuries, many populations have used it to treat gastrointestinal disorders, diabetes, hypertension and inflammation. Scientific studies in recent years have confirmed a high antioxidant and anti-radical activity for *Salicornia* sp.pl., thanks to the high content of flavonoids and polyphenolic compounds which have antidiabetic, neuroprotective, antibacterial, antihypertensive and antitumor activities [83,88].

*A. macrostachyum* could be a potential source of bioactive compounds that are useful for the treatment of several human diseases. It is traditionally used as an antibiotic and as an alexipharmic in Tunisia, and it plays a prominent role in traditional Indian medicine (Ayurveda). This species can tolerate and accumulate heavy metals and has a very high potential for phytoremediation [89]; moreover, the plant biomass has been used for fodder [90].

### 3.3. Soda inermis Fourr. (=Salsola soda L.)

#### 3.3.1. Brief Description, Distribution and Ecology

The genus *Salsola* (from the Latin Salsus, meaning “salty”; Amaranthaceae/Chenopodiaceae family) is widespread in salty areas of arid, semi-arid and temperate regions worldwide, with more than 140 species including both annual and perennial species [91]. *Soda inermis* (=*Salsola soda*; saltwort or barilla plant) is a succulent annual herb 20 to 120 cm high; its succulent leaves are linear, semi-cylindrical, 2–7 cm long and 2 mm wide, and shortly mucronate at the apex, green or reddish color (Figure 8); the flowers are small, sessile, axillary to the leaves, and conspicuously bibracteolate; it has five perianth segments, becoming hardened in fruit. It has wide Eurasian and African distribution and typically grows in correspondence with salt marshes and on soils with high organic content. This species colonizes maritime and inland salt marshes, usually grows in halo-nitrophilous and termophilous vegetation, on soils rich in organic content, and forms floristically very poor (almost mono-specific) communities dominated by S. inermis (*Salsoletum sodae* Pignatti 1953; *Thero-Suaedion splendentis* Br.-Bl. in Br.-Bl. & al. 1952). According to the 92/43/EEC “Habitat” Directive, this vegetation falls within the habitat of community interest 1310—“*Salicornia* and other annuals colonizing mud and sand”.

#### 3.3.2. Food Use

*S. inermis* is appreciated as a vegetable, especially in central Italy, mainly in the Latium region, where it is known as “agretti” because of the slightly bitter taste [92]. It is commonly sautéed with garlic and olive oil and served as a side dish. It can be used in salads, but another more common use is lightly steamed and dressed with lemon juice, olive oil, sea salt and fresh cracked black pepper.

In Salento, it is not collected and used as an edible species, whereas it is used in various localities of Gargano where the young seedlings are eaten both raw and boiled and dressed with vinegar and olive oil [67] (Figure 4; Table 1; Appendix A).

It is interesting to point out that the plant is used similarly, on a small scale, in Japan [93].

#### 3.3.3. Domestication/Cultivation

*S. inermis* is cultivated and used in particular in Italy and in Spain (where it is known as “barilla”). This species can be cultivated in those salty lands where no other crops could give a good yield, or in those areas where irrigation is possible only with salty water. It is spread in south Europe, particularly in marginal areas near the coast [93,94]. *S. inermis* was cultivated also to reclaim brackish swamps [95].

Only for a few years has *S. inermis* been cultivated in Salento and other parts of Apulia. It should be sown in spring. The soil must be well worked and draining, exposed in a sunny position. The sowing must be done in March and the cultivation does not require fertilization. Germination is slow, it takes about a month for all the seeds to germinate and, between May and June, the young plants are ready to be harvested for food consumption.

#### 3.3.4. Other Uses/Properties

In past times *S. inermis* had gained much attention due to its use as soda ash [93]. *S. inermis* is one of those plants that has lost economic importance as a consequence of new industrial processes. In fact, it was formerly highly demanded for the production of impure sodium carbonate that was used, e.g., for the production of soap and glass. This impure salt was called “barilla,” hence the name of the plant. Barilla was obtained from the ashes of *S. inermis* and other halophilous plants able to produce sodium carbonate from the sodium chloride in the soil; it was also obtained from seaweeds. Barilla from *S. inermis* was preferred for the glass industry over that produced from other terrestrial plants because it was richer in potassium salts and therefore preferably employed in soap preparation. Nowadays it is very difficult to give an estimation of the former economic importance of *S. inermis* because soda is also obtained from the ashes of other plants. In fact, the Solvay process, which leads to the rather pure sodium carbonate used in all industrial processes, progressively replaced the production of barilla from plants since the beginning of the present century [95,96]. *S. inermis* is mentioned (as *Salsola soda*) by [97] among the Salento plants useful for obtaining sodium carbonate from the ashes (Appendix A).

*S. inermis* has also played a minor role in popular medicine: it was considered to be diuretic, aperitif and vermifuge [91]. This species can be used for bio-desalination of saline soils and as a companion plant with conventional crops [96,98].

### 3.4. Cakile maritima Scop.

#### 3.4.1. Brief Description, Distribution and Ecology

*Cakile maritima* (sea rocket) is a succulent, annual species belonging to the Brassicaceae family, thriving along Mediterranean and European Atlantic sandy coasts, confined to maritime strandlines on sand or shingle, and associated fore dunes. The species is tolerant of salt spray and temporary seawater inundation. *C. maritima* is a succulent annual herb with prostrate or ascending stem, highly branched, fleshy leaves entire obovate or oblancelate-to-deeply pinnately lobed; its inflorescences are dense, many-flowered racemes, terminating the main stem and branches, with lilac to purple flowers (Figure 9). Together with *Salsola tragus*, it forms the typical vegetation of sandy and shingle beach drift lines (e.g., *Salsolo kali-Cakiletum maritimae* Costa & Mansanet 1981, corr. Rivas-Martínez et al. 1992; *Cakiletea maritimae* Tx. et Preising in Tx. ex Br.-Bl. et Tx. 1952 class). According to the 92/43/EEC “Habitat” Directive, this vegetation falls within the habitat of community interest 1210—“Annual vegetation of drift lines”.

#### 3.4.2. Food Use

Harvested before flowering, the whole plant is boiled for about 15 min. Then drained and passed under hot water in order to eliminate the bitter part (this operation must be repeated 3/4 times). After it can be served with potatoes, stale bread softened in the same cooking water of the plant and then seasoned with olive oil (Margherita di Savoia sandy coast, Foggia).

*C. maritima* is used for food purposes in very few places in Salento (Figure 4; Table 1; Appendix A). The leaves, plump and with a spicy taste, can be added to raw salads and flavor cooked dishes [29].

#### 3.4.3. Domestication/Cultivation

As a therophyte, *C. maritima* can only be multiplied generatively. No cultural protocols are known, nor if the seeds undergo a possible period of dormancy. However, it has been observed that, in conditions of salinity with NaCl equal to 75 mM, there is a significant delay in seed germination and that, in concentrations of NaCl equal to 100 mM, there is a greater vegetative development in the plant, with a greater number and size of leaves, greater water content and greater photosynthetic activity [99]. The response to the progressive salinization of coastal environments in *C. maritima* is immediate and positive, and for this reason it is proposed as a species to be included in agricultural systems as a companion crop or to increase its consumption for food purposes [100]. Several studies, especially from north Africa, have considered *C. maritima* as a promising species for domestication in the context of the biosaline agriculture approach relying on the domestication of halophytes [101,102].

#### 3.4.4. Other Uses/Properties

*C. maritima* is a species of food interest for many Mediterranean populations and represents a promising species, owing its ecological plasticity and economic potential because of its ability to produce numerous secondary compounds and as an oilseed and energy crop. In recent years, its economic interest in its oilseeds and secondary metabolites, which are of therapeutic interest for humans, have been discovered. *C. maritima* has a high content of polyphenolic compounds that perform antioxidant activity and classify it among the species with antiallergenic, anti-inflammatory, anticancer, anticoagulant, antimicrobial, cardioprotective and vasodilatory properties [103,104,105]. In popular medicine *C. maritima* was considered an excellent anti-scurvy and anti-catarrhal, with diuretic and laxative properties, capable of counteracting jaundice and dropsy. In recent years, its microbial and antifungal activity has been evaluated, leading to industrial interest, as a substitute for the traditional antibiotics used for food preservation to which many bacterial and fungal species are now resistant [106].

Furthermore, it has been found that *C. maritima*, within 6 weeks, is able to reduce the phytotoxicity of phenanthrene, a Polycyclic Aromatic Hydrocarbon (PAH), by 75%; for this reason, it is thought to include *C. maritima* among the species to be used in phytoremediation interventions [107].

For all these reasons, the cultivation of this plant on salted marginal soil, in the context of the necessary development of bio-saline agriculture in the future, has been taken into consideration [102].

### 3.5. Crithmum maritimum L.

#### 3.5.1. Brief Description, Distribution and Ecology

Marine fennel (*Crithmum maritimum*, Apiaceae), also known as crest marine, marine fennel, sea fennel, sampier and rock samphire, is a suffruticose chamaephyte, i.e., a perennial plant with woody basal stems and herbaceous upper branches, standing 20–60 cm tall and is glabrous, aromatic and pruinose. It has fleshy leaves, with lanceolate-linear leaflets and inflorescences in umbels with greenish-white flowers (Figure 10a). Its distribution area extents from the Mediterranean and Black Sea coasts, up to the Atlantic coast of Portugal and of south and south-west England, Wales and the Republic of Ireland [108]. This aromatic plant grows wild in rocky coastal environments, such as rock crevices, rocky shores and, sometimes, on shingle beaches. *C. maritimum* typically occurs in the halophytic and halotolerant, perennial plant communities which constitute the first vegetation belts on the rocky coasts in next proximity to the sea, directly exposed to the action of marine aerosol, wind and waves, and tolerating a high concentration of sodium chloride in the substrate (*Crithmo maritimi-Staticion* Molinier 1934; *Crithmo maritimi-Staticetea* Br.-Bl. in Br.-Bl. et al. 1952) [41,51]. According to the 92/43/EEC “Habitat” Directive, this vegetation falls within the habitat of community interest 1240—“Vegetated sea cliffs of the Mediterranean coasts with endemic *Limonium* spp.”.

#### 3.5.2. Food Use

Especially in the coastal villages of Salento, it is a tasty ingredient used to flavor fish-based dishes and for the preparation of many side dishes. The tender apical portions of the stems are harvested at the end of spring, peeled, blanched, drained and then seasoned with garlic and mint for long-term potting in oil or vinegar. In recent years, someone has experimented with freezing the freshly blanched plant, to be thawed and seasoned as a side dish (with oil, vinegar, garlic, mint and breadcrumbs (Figure 10b) or to be used as a flavoring [65,68]. In other parts of the Apulia region, fresh leaves are julienne cut, seasoned with oil and lemon (or vinegar) and consumed with salads; whole leaves can be used to make omelettes (Figure 4; Table 1; Appendix A).

Since ancient times, the fresh consumption of sea fennel has been recommended for its purifying, tonic, diuretic and purgative properties; the fruits of sea fennel were used in ancient times for reducing fermentation and intestinal spasms. Rich in vitamin C, sea fennel was also used as an anti-scurvy by sailors.

For promoting a full exploitation of this species, a new food product was obtained by drying sea fennel using different treatments (air-drying, microwave-drying, microwave-assisted air-drying and freeze-drying) [109]. Water activity, essential oil content, chlorophylls, surface color, coloring power and sensory evaluation were analyzed; the results indicated that microwaving and freeze-drying are optimal for preserving qualitative traits, including organoleptic properties, in dried sea fennel for food use. The culinary use of the sea fennel for several gastronomy products as a new spice-colorant has been also reported by [92].

#### 3.5.3. Domestication/Cultivation

Marine fennel is cultivated in many areas across Europe for several economic and industrial purposes; in general, sea fennel is a strictly heliophile and therefore requires a planting pattern that ensures adequate inter-and intra-row spacing to maximize the leaf area exposed to the sun [106,108,110,111]. 

Despite this, species usually grow in close proximity to the seawater, salinities exceeding 50 mM NaCl were found to inhibit its germination [112,113]. According to [113], a useful approach to overcome the salt-induced seed dormancy observed in halophytes consists in the exogenous application of germination-promoting substances; in this way, nitrate, ammonium, and GA3 proved to significantly enhance seed germination of *C. maritimum* under salinities. Interestingly, red light application was also efficient for seed germination induction under salinity.

Propagation by softwood cuttings does not seem to be the appropriate method for mass propagation, as the mother plants of sea fennel provide a limited number of cuttings. In vitro culture techniques have been extensively used not only for rapid clonal propagation but also for the study of the salt tolerance mechanisms of many species; a first study on *C. maritimum* has been reported by [114].

In the Botanical Garden of the University of Salento, the multiplication of sea fennel has been carried out to be used for living collections and for interventions of environmental restoration on coastal habitats. The propagation by the vegetative method has not been successful, as the stem and the young branches have no rooting power, as said just above; on the other hand, the propagation by the generative method has provided an excellent result, certifying the germinative power around 70–75% and without germination-promoting substances. In this experiment, seeds were harvested between October and November, cleaned of impurities and stored in paper bags; then, they were sown in the second ten days of December on a substrate consisting of a mixture of peat and river sand in a 1:1 ratio, in honeycombed plateau placed in a cold tunnel. The emergence raised one month later, characterized by a slow scaling. In the plateau, the complete development of the root system took place after 5 months. Young plants tolerated well both transferring to larger pots and transplanting in the open field at the end of spring or in full autumn.

In the context of the use of sea fennel for the creation of green roofs, specific cultivation protocols have been provided. According to [115], sea fennel plants were satisfactorily grown on a soilless substrate, a mixture of grape marc compost, perlite and pumice, with a shallow depth of 15 cm. The better growth of sea fennel plants on a substrate consisting of pumice, perlite, compost, peat and zeolite, with a depth of 15 cm compared to a depth of 7.5 cm, is also reported by [116], while the same researchers found that the growth of plants was greater in the deficit irrigation treatment of 60% evapotranspiration (ETc) compared to the irrigation treatment of 30% ETc.

The use of *C. maritimum* in agricultural practices is an interesting possibility in areas where the growth of other species is limited. An evaluation of different agronomic protocols that have previously not been investigated for the cultivation of this species was proposed by [117].

#### 3.5.4. Other Uses/Properties

In recent years, new scientific research has revived the interest in sea fennel, not only as a food plant due to its nutritional value, but also as a species of pharmacological interest [118,119]. *C. maritimum* is rich in phyto-compounds to which its numerous uses in medicine are owed. Roots, leaves and fruits are rich in several bioactive substances (essential oils, iodine, trace elements, beta carotene, proteins and mineral salts) with a wide range of uses as aromatic, medicinal, antimicrobial and insecticide. A large number of scientific studies show its effectiveness as an appetizer, tonic, digestive, carminative, diuretic, vermifuge and antimicrobial, as well as to treat kidney and heart disease [120,121,122,123]. Moreover, it has antiscorbutic properties, owing to the vitamin C content [124]. Recent studies have highlighted its properties as antioxidant, vasodilator, antibacterial, cytotoxic, cholinesterase inhibitory and anticancer [125,126,127,128,129]. Dehydrated and pulverized, it is an excellent flavoring and coloring agent for foods [92]. It was once used for the production of soda [110]. In temperate climates, the plant is used for ornamental decoration in rock gardens along the sea [129]. As a species that also tolerates strong insolation and long periods of drought, it is recommended in the construction of green roofs in the context of urban horticulture [115,116,130].

### 3.6. Reichardia picroides (L.) Roth

#### 3.6.1. Brief Description, Distribution and Ecology

*Reichardia picroides* (Asteraceae) is a scapose hemicryptophyte, 20 to 40 cm long, with lush-green leaves spatulate to oblanceolate, entire to pennatopartite, gathered in a basal rosette; the branches of the flowering scape terminate with capitula of yellow ligulate flowers (Figure 11a,b). This species has a strictly Mediterranean distribution; it is a salt-tolerant species, with effective adaptation mechanism against saline conditions [131]. The var. *maritima* (Boiss) Fiori, no longer recognized as a valid taxon, with fleshy leaves and typically growing in coastal rocky habitats, has been considered as a diagnostic species of the *Crithmo-Staticetea* class [41,50]. According to the 92/43/EEC “Habitat” Directive, this vegetation falls within the habitat of community interest 1240—“Vegetated sea cliffs of the Mediterranean coasts with endemic *Limonium* spp.”.

#### 3.6.2. Food Use

*R. picroides* is among the best-known edible species in Italy, present both in coastal and inland habitats [132]. It rarely forms extensive and dense populations, and this makes its research for the collection more careful. In Puglia, it is used as only ingredient in soups. Generally, it is boiled and then dressed with olive oil, or sautéed in oil with onion and then enriched with cheese and other ingredients. In Salento, added to other wild herbs, it makes the “fojemmìsche” even more pleasant; in Martano (LE), it is referred to with the dialectal name “cannazzicula” to indicate its ability to delight the palate [65]. It has a sweet and delicate taste and can also be eaten raw in salads (Gargano) (Figure 4; Table 1; Appendix A). From November to July, the plant is always available; the basal rosette is collected, and the internal leaves ore those more edible. The still-closed flower heads can be used to flavor and decorate dishes or to add to salads. When the plant is old, it has a pungent taste, hence the epithet “picroides”. 

In the Umbria Region (Central Italy), its roots are roasted as a coffee substitute [133].

#### 3.6.3. Domestication/Cultivation

This species is widely used as a wild vegetable and there is a hypothesis about some attempt at cultivation in the past [134]. Nevertheless, at present, no multiplication experiences by agronomic cultivation of *R. picroides* are reported.

Rich in phenolic compounds and tocopherols, the species can be cultivated under unfavorable conditions that can improve the bioactive properties through the increased phytochemicals content [24,131]). In a recent study [135], *R. picroides* was grown for four or six weeks under a greenhouse in a floating system. In order to improve the nutraceutical quality of the tissues, the plants were exposed to the following NaCl concentrations: 1.7 (control), 25, 50 and 100 mM. The results showed that a 4-week growing period in a floating system with 50 mM NaCl increased the content of bioactive molecules without affecting the fresh yield. After six weeks of cultivation, despite a decrease in biomass production as compared with the control, the leaves of salt-treated plants contained higher levels of bioactive molecules along with lower amounts of nitrate ion.

#### 3.6.4. Other Uses/Properties

According to the popular pharmacopoeia of many countries, *R. picroides* has hypoglycaemic, diuretic, depurative, galactogenic and tonic properties [136,137]. In Sardinia, it was even used as a popular treatment against heart diseases such as angina pectoris [138]. Recent studies have confirmed that *R. picroides* extract has high antioxidant activities (at doses lower than 250 mg/kg) thanks to a high content of phenols and can have many therapeutic applications [30,139]. The ability of *R. picroides* extract to inhibit postprandial platelet aggregation in vitro has also been demonstrated [140].

*R. picroides* is also used for feeding rabbits in some areas of the Basilicata region (southern Italy) [141].

### 3.7. Silene vulgaris (Moench) Garcke subsp. tenoreana (Colla) Soldano & F. Conti

#### 3.7.1. Brief Description, Distribution and Ecology 

*Silene vulgaris* (Caryophyllaceae) is native to Eurasia. In Italy, it is present with six subspecies [69]. The subspecies differ from each other in their vegetative habit, leaf shape, leaf size, etc. The subspecies vulgaris is the most common and it grows in all Italian territory except in arid zones, and ranges from 0 to 1500 m asl (rarely up to 2400 m asl) [70]. It is a glabrous or poorly pubescent perennial herb, 30–70 cm high, generally with erect habit but sometimes prostrate; its leaves always sessile and opposed, the largest ones (the median) 12–18 mm × 40–60 mm are linear-lanceolate acute but not pointed; it has 3–9 flowers per plant, gathered in bunches, pendent on flexuous peduncles 5–15 cm long; it has an ovoid calyx (twice longer than wide), much wider than capsule, therefore apparently inflated around them; it has five white or lightly rosy petals with nails as long as the calyx; its anthers and style are purple; its capsule three times longer than the carpophore [70].

*S. vulgaris* subsp. *tenoreana* is similar to the related *S. vulgaris* subsp. *vulgaris*, but with a woody stem at the base and narrower, fleshy leaves. This taxon has an eastern Mediterranean distribution, preferentially growing in coastal areas, on rocky substrates (Figure 12), but also inland thanks to its great adaptability. It frequently appears, as “companion” species, in the communities of the *Crithmo-Staticetea* [41]. According to the 92/43/EEC “Habitat” Directive, this vegetation falls within the habitat of community interest 1240—“Vegetated sea cliffs of the Mediterranean coasts with endemic *Limonium* spp.”. A recent study carried out to explain the current distribution of *S. vulgaris* on the European territory has shown that the species had a post-glacial distribution towards the North Pole ice cap from the refugia of southern Europe, contrary to the hypothesis that it has undergone a westward migration during agricultural expansion, such as most of the wild herbs that accompanied agricultural crops [142].

#### 3.7.2. Food Use

The tender leaves and the apical parts of the young stems are harvested, which are already abundant after the first autumn rains and in the spring. In Salento, numerous recipes mention it as a single ingredient (risotto with “strigoli”, green pasta with “strigoli”, sautéed “strigoli for filling focaccia”, omelette with “strigoli”, salads) [29], but it is also an ingredient in the so called “fojemmìsche”. In the area of Monopoli (Bari), the leaves are cooked as part of an omelette with eggs and cheese (Figure 4; Table 1; Appendix A).

It is also culinarily used in other regions of Italy: the leaves are cooked and stewed in Amalfi coast (Campania Region) and the same use is reported in Tuscany [136], in the Marche region [143] and in Sicily [144]. In Veneto [145], its leaves are added to vegetable soups.

The subsp. *vulgaris* is also widely used in Italy as a vegetable and consumed in similar ways to subsp. *tenoreana*. In several localities of southern Italy, *S. vulgaris* subsp. *vulgaris* (named “culicid” in Apulia) are used for their more tender leaves which are fried with olive oil and eggs for preparing a sort of omelette. Shoots are in great demand by gourmets who use them for preparing vegetable soups, rice soups, minestrone or risotti. This is also true in Austria, Switzerland and Germany where folk names of the species often contain elements like “kohl” (cabbage) or “spinat” (spinach) [146,147].

#### 3.7.3. Domestication/Cultivation

It is a rustic and very vigorous plant; it grows very well on any type of soil; exposure should be sunny. Generative propagation is possible. The seeds should be collected between June and July and selected, cleaned and stored in paper bags. Among the propagation techniques used at the Botanical Garden of Lecce, direct sowing on loose substrate (seed should be scattered sparsely by broadcasting or in rows 30–35 cm apart) enriched with organic fertilizer was adopted.

The case of *S. vulgaris* is an example of domestication processes from use to cultivation which is characteristic for many other vegetable plants in Italy [93,148] and which continues at the present time.

#### 3.7.4. Other Uses/Properties

*S. vulgaris* has played a minor role in popular medicine: its rhizome contains saponin, alkaloids and tannins and it is considered to be a depurative of blood and anti-anaemic; its cultivation for producing those substances was attempted in southern Kazakhstan [93]. 

*S. vulgaris* is rich in mineral salts, and is therefore considered an excellent tonic [60,61]. This species is capable of colonizing contaminated and, because of its capacity to retain mercury and heavy metals, could be used in phytoremediation technologies [149,150].

### 3.8. Allium commutatum Guss

#### 3.8.1. Brief Description, Distribution and Ecology

*Allium commutatum* (Amaryllidaceae), commonly named maritime wild leek, is a taxonomic entity typical of coastal rocky environments with central-eastern Mediterranean distribution. It has bulbous geophytes (bulbs are halo-tolerant) up to 100 cm tall and more, with showy spherical inflorescences (Figure 13a). It is usually found in open, rocky and/or stony places near the sea (Figure 13b), but also in fallow or abandoned fields and marginal areas. It grows from sea level to about 300 m asl and flowers from the end of June until the end of July [151]. In Italy, it is very common in coastal areas in the south (including Sardinia and Sicily), with the northern limit at the Tuscan archipelago (Tyrrhenian) and Marche (Adriatic). It is diagnostic species of the *Crithmo-Staticetea* class [41], that is, according to the 92/43/EEC “Habitat” Directive, habitat of community interest 1240—“Vegetated sea cliffs of the Mediterranean coasts with endemic *Limonium* spp.”.

#### 3.8.2. Food Use

*A. commutatum* is a primary wild relative of and potential gene donor to leek (*A. porrum* L.) and a tertiary wild relative of and potential gene donor to a number of other crops in the *Allium* group, including onion (*A. cepa*), Welsh onion (*A. fistulosum*), garlic (*A. sativum*) and chives (*A. schoenoprasum*) [152,153].

As mentioned above, maritime wild leek is quite frequent on the Apulian coasts, especially on rocky coasts. As many other *Allium* species found in the inland area (*A. sativum* L., *A. vineale* L., *A. ampeloprasum* L., *A sphaerocephalon* L., *A. roseum* L., etc.), maritime wild leek is edible although not much used, probably because other wild and cultivated species are already abundantly available in the area. In Mediterranean and southern Italian cuisines, garlic is the main flavoring ingredient of numerous dishes: meat stews; a flavor enhancer for fish and grilled vegetables; greens and vegetables seasoned with mint and garlic before being preserved in oil or vinegar; marinades with breadcrumbs, vinegar, saffron, mint and garlic; oil flavored with garlic cloves [29,60,65] (Figure 4, Table 1).

The most commonly used parts of the plant are the bulbs, usually consumed raw or cooked. In the Gargano area, they is consumed raw in mixed salads or roasted and dressed with olive oil [66]; moreover, young leaves are used to flavor salads and soups and young seedlings are harvested and roasted. It is also used in sautéed foods, which are the basis of sauces, to season “friselle” (Monopoli, Bari) (Appendix A). 

In traditional Sicilian cooking, a special dish, in the winter, is represented by the fried bulbs (the so called “purrietti”) [154].

The consumption of the bulbs, eaten with bread, has been reported for various locations in eastern Mediterranean [155].

#### 3.8.3. Domestication/Cultivation

*A. commutatum* can be propagated both by seeds and generatively. The ripe inflorescences, with the capsules containing mature seeds, are harvested between June and July; the sowing is to be done between October and November on a loose and draining substratum. Propagation by vegetative means involves the detachment of the largest bulbils from the main bulb and their transplantation into deep soil at the end of autumn.

Research on shoot and root regeneration from callus tissue of *A. commutatum* is reported by [156].

#### 3.8.4. Other Uses/Properties

Like other *Allium* species, *A. commutatum* has numerous therapeutic properties. It can be used raw, in decoction or infusion for intestinal infections, for respiratory system diseases as an expectorant and anti-catarrhal, antibacterial, anthelmintic, hypotensive, antirheumatic and anti-inflammatory. Garlic poultices can be applied on inflamed or burned skin, on limbs and parts affected by bursitis, rheumatism and other strong inflammations of the tissues. For the populations settled between the Tigris and the Euphrates, a panacea for all ills was the decoction prepared with wine, garlic and calamus seeds. In addition to the nutraceutical and medicinal value, garlic was attributed a poisonous power against demons, evil spirits, the evil eye and witches [29,60,65].

Recent studies have evaluated the chemical profile of *A. commutatum* obtaining an alliin content of 31.5 mg/g in ethanolic extracts of bulbs (BE) and 38.8 mg/g in ethanolic extracts of aerial parts (APE); in the latter, quercetin (38.5 mg/g) and luteolin (31.8 mg/g) were also present. The bulbs and leaves are a precious source of organic compounds that fight obesity and perform antioxidant activities [157].

In other species of the *Allium* genus, studies of organic compounds confirm antioxidant, antiseptic and anticancer activities thanks to the high content of reduced glutathione, flavonoids, soluble proteins, vitamin C, carotenoids, chlorophylls a and b and malonyldialdehyde [158,159].

### 3.9. Beta vulgaris L. subsp. maritima (L.) Arcang

#### 3.9.1. Brief Description, Distribution and Ecology

*Beta vulgaris* subsp. *maritima* (sea beet or wild beet) is a plant species belonging to the Amaranthaceae/Chenopodiaceae family (subfamily *Betoideae*). Scapose chamaephyte, with a dense basal rosette of leaves of spatulate shape (but variable in shape and size), is shiny green with inconspicuous greenish flowers in groups of two-to-five flowers in long spikes (Figure 14a). It has a wide distribution ranging from the Canary Islands in the west, northward along Europe’s Atlantic coast and Baltic Seas, extending eastward through the Mediterranean basin up to the coasts of the Black Sea. It usually grows on stony and pebbly soils along the coasts but also, more rarely, inland on clays; it is commonly cultivated from sea level to about 600 m. This species has a large environmental adaptability to conditions such as high salinity and poor soil, which is related to its extreme genotypic and phenotypic variation [160].

#### 3.9.2. Food Use

Sea beet is one of the most appreciated wild herbs for the preparation of typical dishes. It is abundantly available not only in coastal environments but also in those immediately adjacent to. The tender leaf lamina is scarcely fibrous, and, after a very short blanching, it is pounded and added to flour mixture to make the so called “green pasta”. In Salento, the sea beet is a fundamental ingredient for the “fojemmische”, a typical dish which has as ingredients a mixture of 10–20 wild herbs, depending on the season. Used as a single ingredient in almost all the sited visited in the Apulia region (Figure 4; Table 1; Appendix A), the methods of preparation are simple: boiled and seasoned with oil and lemon (Figure 14b) or sautéed with onion and then enriched with pancetta, speck, ricotta, cheese, to be used as a filling for focaccia [61,161]. It is also commonly used in broad bean and legume soups, or in the preparation of omelettes. In the Gargano area, it is used to enrich a local dish called “pancotto”, a dish in which the plant is boiled and with the addition of tomatoes and spices and the use of stale bread, was a typical dish in the tradition of the peasant people.

In Sicily, traditionally, in order to avoid constipation, it is recommended to consume the leaves of sea beet (locally called “gira”), which are also considered refreshing [162].

#### 3.9.3. Domestication/Cultivation

Sea beet is considered the progenitor of all the crop varieties of beet [64,163,164,165]. Its domestication is traced back to the Egyptians and Babylonians. The Greeks and Romans made extensive use of it and selected a large number of varieties handed down to the present and all with an annual or biennial cycle. 

Sea beet is a perennial species; propagation by the vegetative method is not known whilst its propagation by the generative method is possible. The seeds are difficult to release because they are enclosed inside hard glomeruli which are soaked for 24 h and then sown on a soft and draining substrate.

Sea beet survives in extreme conditions, from brackish marshes to cliffs continually beaten by the waves; it has the ability to tolerate high salt concentrations and aridity, unlike all the other cultivated varieties which instead show signs of suffering and morpho-functional changes such as growth retardation, wilting of the leaves, reduction of photosynthetic speed and stomatal conductance [166].

#### 3.9.4. Other Uses/Properties

The food value of sea beet is to be attributed to the high content of fibers, mineral salts, vitamins (A, B1, B2, PP, C) and proteins that provide disinfectant, diuretic, refreshing, purifying and laxative and anti-anemic properties [60,65]. In the first century BC, the Greek physician and botanist Dioscorides proposed using sea beet against earache, against lice and dandruff and to apply boiled root on pustules and burns [163]. An essential oil was distilled from the aerial part of sea chard which revealed antioxidant, anticholinesterase, anti-tyrosinase and cytotoxic properties on the A549 cell line [167]. A recent study showed that the leaves’ extracts are usefulness to prevent diabetes complications and have promising chemo-preventive properties [23].

### 3.10. Capparis spinosa L.

#### 3.10.1. Brief Description, Distribution and Ecology

*Capparis spinosa* (Capparaceae), commonly known as flinders rose or caper bush, is a procumbent shrub with semi-prostrate (sometimes pendulous) unramified (or scarcely branching) branches, 40 to 80 cm long, green sometimes to reddish or yellowish. Leaf stipules may be formed into spines, granting it the name “spinosa”. Its leaves are usually rounded to ovate; its flowers are solitary, with four white-pinkish obovate or rounded-ovate petals, and numerous stamens (Figure 15a,b). The fruits are like small cucumbers when immature, and when ripe they enlarge and turn purplish-red-brown with numerous seeds plunged into a white mucilaginous placenta. The species is native to dry regions of western and central Asia; at present, it grows naturally from the Atlantic coast of the Canary Islands and Morocco to the Black Sea and to the east side of the Caspian Sea; it is also spread in north Africa, Europe, west Asia, and Australia [168,169]. It requires average annual temperatures above 14 °C and average annual rainfall of no less than 200 mm; moreover, it resists strong winds [170]. The drought and salt tolerance of *C. spinosa* allows it to persist in a wide range of habitats, even on nutrient-poor, rocky and gravelly soils [170]. The species has a wide distribution in anthropogenic wall and rock crevices chasmophyte nitrophilous vegetation (*Cymbalario-Parietarietea diffusae* Oberdorfer 1969 = *Parietarietea judaicae* Oberdorfer 1977) of both coastal and inland areas.

#### 3.10.2. Food Use

The consumption of the caper buds and fruits was well known already in Greek and Roman times, both as a condiment and for therapeutic use, and for this reason caper cultivation was widespread in the conquered territories. Commonly, the closed buds (capers) are harvested for use and consumption, but not too far into their development because when small they are crunchier and tastier. Capers can be kept for long periods, even 2–3 years if they are harvested, put immediately in a glass jar and covered with vinegar, or made into a pickle (that is covered with coarse salt and turned over for 4–5 days, then put into jars and covered with the same salt; Figure 15c). In some localities of Salento, the tender tips of the stems with the last two-to-three very tender leaves are also collected and transformed, with the same dialectal name that is given to the “chiapparara” plant; however, the removal of the apex implies the arrest of the elongation of the stem, and for this reason the producers avoid doing it. Capers are used as a condiment in raw salads, in caponata, with fish, in fillings and in sauces. Throughout the Apulia (Figure 4; Table 1; Appendix A), they are essential ingredients of the characteristic “rustic pizza”, stuffed with onions, black olives, tomatoes and capers [65]. The bibliographic sources do not refer to the use of freshly picked capers, but only after maceration in vinegar or pickle, probably due to the strong bitter taste of the raw buds.

The plant starts producing flower buds in the month of June and continues until the month of August by lengthening the stems, on which ripening fruit, fruit setting and new flowers successively appear. The aerial part loses its leaves and dries between October–November; the growing season re-starts in February–March.

#### 3.10.3. Domestication/Cultivation

*C. spinosa* has always been considered of minor importance among the plants of agricultural interest. However, on the island of Pantelleria and in the Aeolian archipelago (and, in particular, on the island of Salina), the caper has reached an increasing economic importance, which has led to the developing of specialized cultivation [171,172]. It is also widely diffused in cultivated form, especially in Southern Europe (Italy, Spain and Greece), in North Africa (Tunisia, Morocco and Egypt) and in the Middle East (Syria and Turkey) [173]. In Linosa (Sicily), island cultivation practices provide rooted cuttings placed in holes 30 cm deep with blond peat to increase soil water-holding capacity. During the first year of growth, five rescue irrigations were carried out in summer to encourage establishment of the young plantings. Pruning was carried out at the end of each year during the autumn–winter period (November–December) by cutting branches to approximately 6–10 cm from the base (long pruning). Subsequently, three-to-four lava stones were placed around the plantings to protect them from the wind and to limit water loss from evaporation; crop care included manual weeding five times and hoeing three times [174]. Additional information on the cultivation of capers in the Pantelleria and Pelagie archipelagos is reported by [175].

*C. spinosa* is a dominant plant element on the limestone cliffs of the Apulian coast. It adapts very easily to harsh environments, from the coastal cliffs to the urban walls and monuments. The root system of *C. spinosa* creeps into the rock fissures and develops to make up 62.5% of the total plant biomass already 4–5 months after germination; the cortical layers of the roots, both fictive and fibrous, are able to store large quantities of water and to overcome water crises [176].

Its seeds have low germinative power and, for this reason, various pre-germinative treatments are used in generative propagation: mechanical scarification of the integuments, cold stratification, immersion of the seeds in a 0.2% H_2_SO_4_ solution and seed treatment with gibberellins (G4+7 and GA3). The vegetative propagation consists of the realization of woody or semi-woody cuttings, 15–30 cm long, with a diameter of 1–2.5 cm, to be taken between February and March, then stratified in sand and placed in a cold box at an ambient temperature of 3–4 °C [169,177].

Propagation experiments carried out in the Botanical Garden of Lecce have shown a good germination percentage (75–80%) for seeds of *C. spinosa* collected in Marina Serra di Tricase (LE) in the month of October and sown in the second ten days of December in an alveolar plateau (kept in a cold tunnel), on a substrate made up of a mix of peat and agri-perlite in a 1:1 ratio. No treatments were carried out to accelerate or improve germination, which began 40 days after sowing, continuing until the month of May.

#### 3.10.4. Other Uses/Properties

*C. spinosa* has a long history as an archaeophyte. This species has been used in medicine since ancient times as testified by the Bible and the writings of Hippocrates, Aristotle and Pliny the Elder. Its numerous pharmacological effects are due to the richness of secondary metabolites. Traditional medicines have handed down different uses: in Iran, traditional medicine recommended the bark of the roots and fruits against malaria, hemorrhoids and as a diuretic; in Pakistan, the plant and roots were the parts used for the preparation of decoctions or infusions with purifying and analgesic power to relieve rheumatic pains, toothaches and coughs; in China, it was used against dropsy, anemia and stomach pain; in Morocco, it is recommended for conjunctivitis, for the expulsion of kidney stones, for diabetes and for gastrointestinal disorders [25,169,178]. Recent scientific studies attest to the presence of very important phytochemicals such as phenols, flavonoids, tocopherols, terpenes which attribute important therapeutic effects to the caper: antihypertensive, antiobesity, anti-inflammatory, hypoglycemic, antibiotic, anti-allergic and antihistamine, hepatoprotective and antitumor. Almost all parts of the caper plant contain high quantities of glucosinolates (84–89%). Glucocapperin (methyl glucosinolate) is present above all in the buds and shoots and glucocleomine is present in the seeds and leaves. In the lipid fraction of the seeds, a high content of linoleic and oleic acid, sterols and tocopherols, aliphatic and triterpenic alcohol was detected [179,180,181,182,183].

No acute or chronic toxicity has been detected for the use of *C. spinosa* and/or its extracts [179,184,185,186].

The people of Israel produced wine from flower buds, and one of the uses of this wine was for preparing incense for the Temple [187].

*C. spinosa* is used as a fodder and ornamental plant, too [186].

This plant, thanks to its drought-resistance and strong root system, has high potential against desertification and soil erosion [169].

## 4. Conclusions

Coastal areas have an important ecological value, and are among the most threatened environments, both in the Mediterranean region and worldwide. In fact, especially in recent decades, they are undergoing rapid anthropogenic development. Increasing human pressure (e.g., urbanization, exploitation of natural resources, plant invasion) is causing degradation of coastal areas, along with the reduction, fragmentation and isolation of their habitats [57,188]. Land claims, agricultural intensification and hydrological modifications are the main drivers of changes [189,190,191,192]. Marine and coastal ecosystems provide a wide range of services to human society including supporting, regulating, cultural and provisioning services; among the provisioning services, the provisioning of cultivated crops (“nutrition”) and of genetic materials (“materials”), especially of genetic resources for new crops, are included [193,194,195,196].

The changing climate by global warming and the increasing aridity, along with sea-level rising, contribute to the salinization of the hydromorphic soils, especially in coastal areas, and determining, among others, serious environmental hazards in agriculture with which Mediterranean policymakers and scientists are beginning to interact through integrated initiatives such as REstoration ACTions for the MEDiterranean (REACT4MED) (https://react4med.eu, accessed on 30 November 2022). In this framework, and in view of the subtropical conditions to which many territories in the Mediterranean region (and, among these, many areas in southern Italy) are approaching due to climate change, halophytes become a valid alternative to conventional vegetable crops, and a strong point for agricultural reconversion and regeneration programs, as they offer the possibility of selecting those more tolerant to dry farming and saline stress, and those which under stress conditions produce greater amounts of secondary metabolites useful for human health [86,167,197]. Focusing on the Apulia region, in Salento, the problem of salinization of groundwater is becoming increasingly serious, especially in coastal areas due to the ingression of sea water. Therefore, halophytes may be used as both associated crops and for human consumption. Moreover, the growing halophytes can also be used in phytoremediation processes of soils polluted by heavy metals or polycyclic aromatic hydrocarbons.

In the framework of the Integrated Coastal Zone Management (ICZM) [198], the creation of buffer zones surrounding protected and sensitive coastal sites, such as those in spatial contact to agricultural areas subject to intensive exploitation, may contribute to both mitigate the effects of agricultural practices on coastal environments and recovery of marginal areas [199]; the latter may be abandoned lands due to soil salinization, which could be subject to dry farming with halophyte cash crops. This practice, apart from preventing possible conflicts with stakeholders (e.g., farmers) standing in or around the protected areas, could also mitigate the effects of gathering wild halophyte species; in fact, as highlighted in the Results, many of the considered species in the wild fall within vegetation types that are protected by the EU Directive as habitat of community interest and, in those places where gathering occurs extensively, natural populations, along with habitat structure, may be put at risk.

Among the most recently interesting cases, *Salicornia* sp. pl. is a good candidate for dryland farming in the presence of salinized soils, and also for reclamation of barren lands and salt flats; to be an economically viable cash crop, this emerging crop should ensure high-yield production [77,200]. Current knowledge suggests that sea fennel also has good potential as an emerging cash crop, even in the context of a saline agriculture regime; nevertheless, this halophyte plant is underutilized for commercial cultivation, possibly by the lack of consumer demand [109,129].

For a full exploitation of these species, further studies on domestication and cultivation practices, as well as on possible product transformation, are needed. Moreover, an overall promotion campaign of these products and their nutritional and healthy virtues should be planned, also in terms of new food products.

## Figures and Tables

**Figure 1 plants-12-00549-f001:**
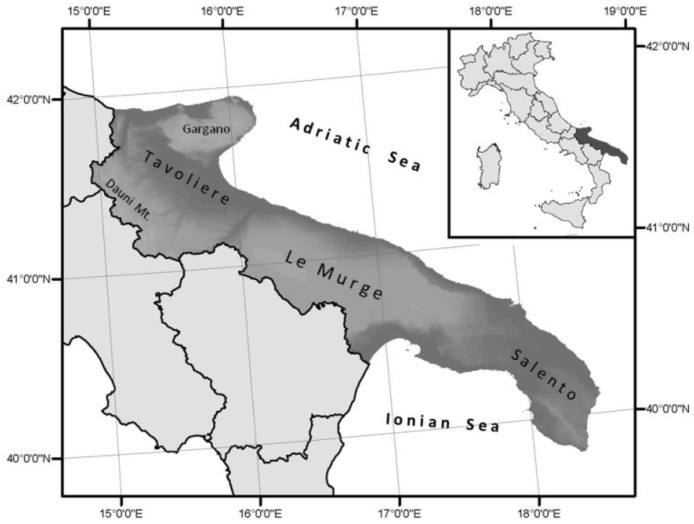
Study area.

**Figure 2 plants-12-00549-f002:**
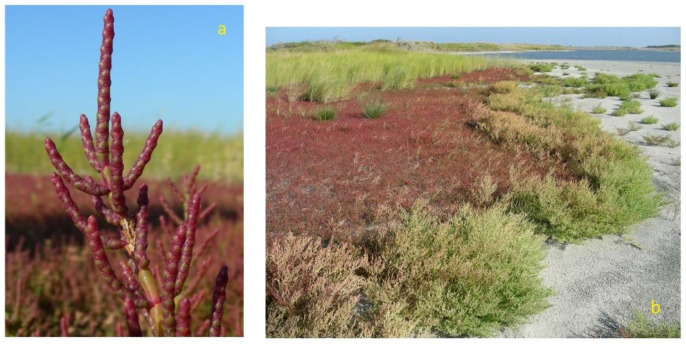
A specimen of *Salicornia patula* (**a**); *S. patula* community ((**b**); the reddish glasswort carpet) at the edge of a coastal lagoon in Salento.

**Figure 3 plants-12-00549-f003:**
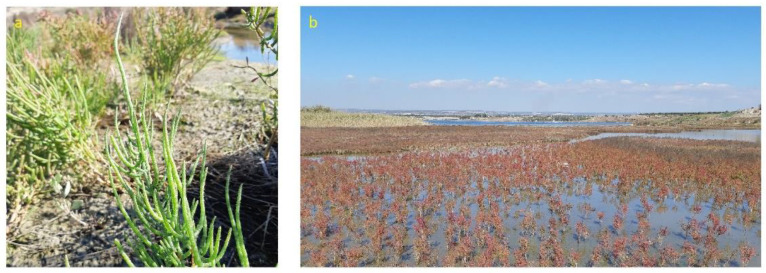
*Salicornia dolichostachya* (**a**); a plant community dominated by *S. dolichostachya* and *S. emerici* (**b**) at “Palude La Vela”, Taranto, southern Apulia).

**Figure 4 plants-12-00549-f004:**
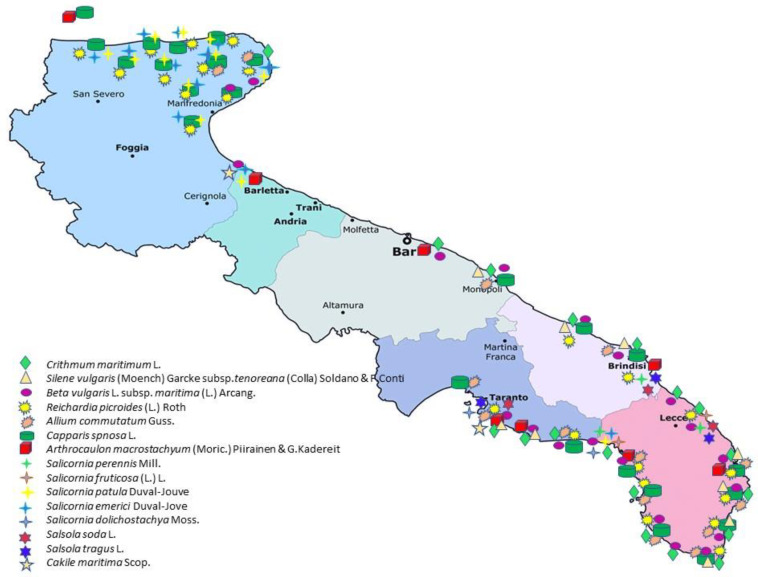
Distribution, across the Apulia region, of the food use of halophytes and halotolerant species treated in this contribution, according to our field observations and literature data.

**Figure 5 plants-12-00549-f005:**
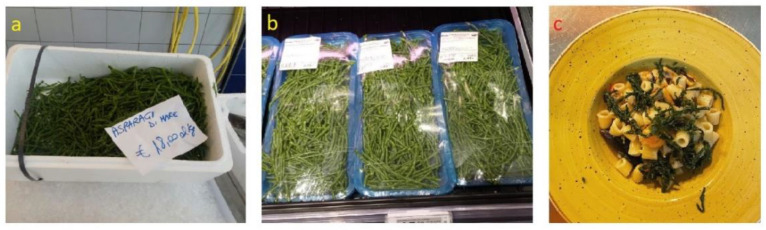
*Salicornia* sp. sold at a fish shop (**a**) and at a superstore (**b**) in Bari; a dish of pasta with *Salicornia* sp. (**c**); Peschici, Gargano, Northern Apulia).

**Figure 6 plants-12-00549-f006:**
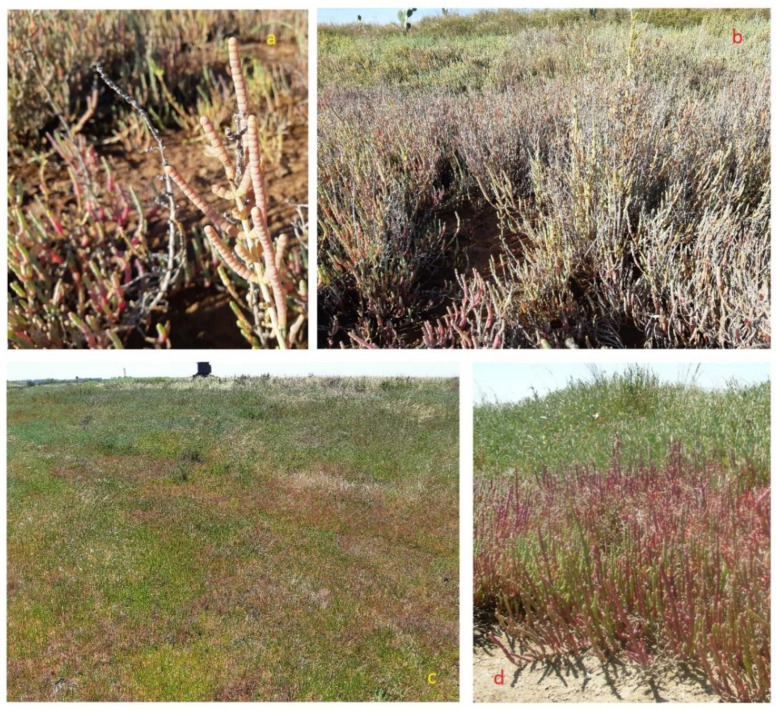
*Salicornia fruticosa* (**a**,**b**) at “Saline Margherita di Savoia” (FG, Northern Apulia); *S. alpini* (**c**,**d**) at “Saline di Punta della Contessa” (BR, Southern Apulia).

**Figure 7 plants-12-00549-f007:**
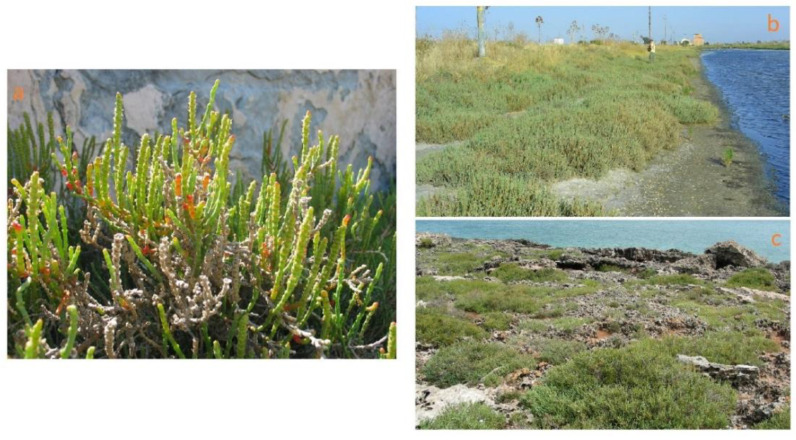
*Arthrocaulon macrostachyum* (**a**); *A. macrostachyum* communities at the edges of the “Capitanata” coastal lagoons ((**b**); Northern Apulia) and on the rocky coasts of Torre Guaceto ((**c**); Salento, Southern Apulia).

**Figure 8 plants-12-00549-f008:**
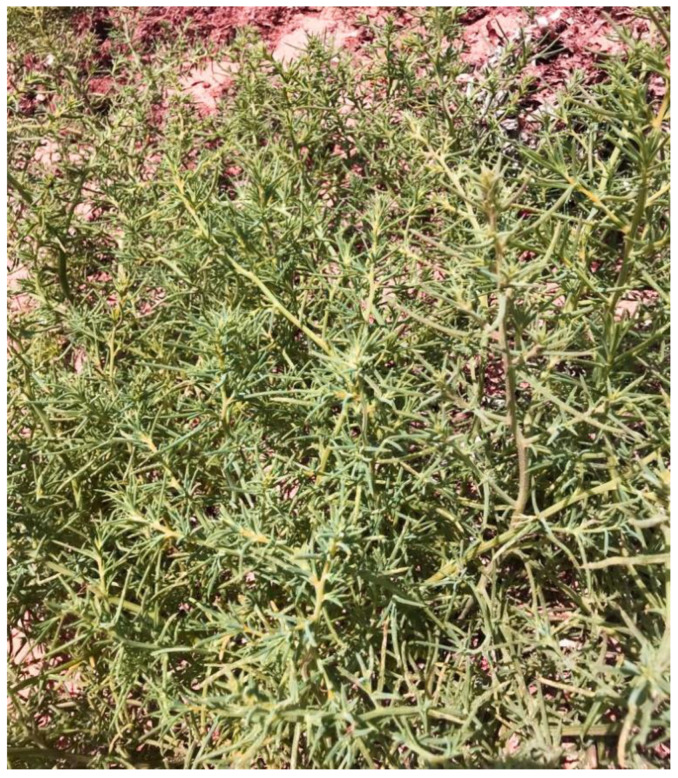
*Soda inermis*.

**Figure 9 plants-12-00549-f009:**
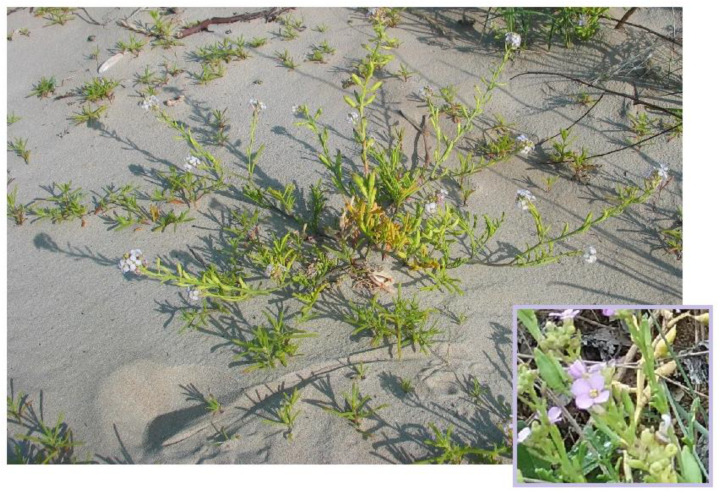
*Cakile maritima*.

**Figure 10 plants-12-00549-f010:**
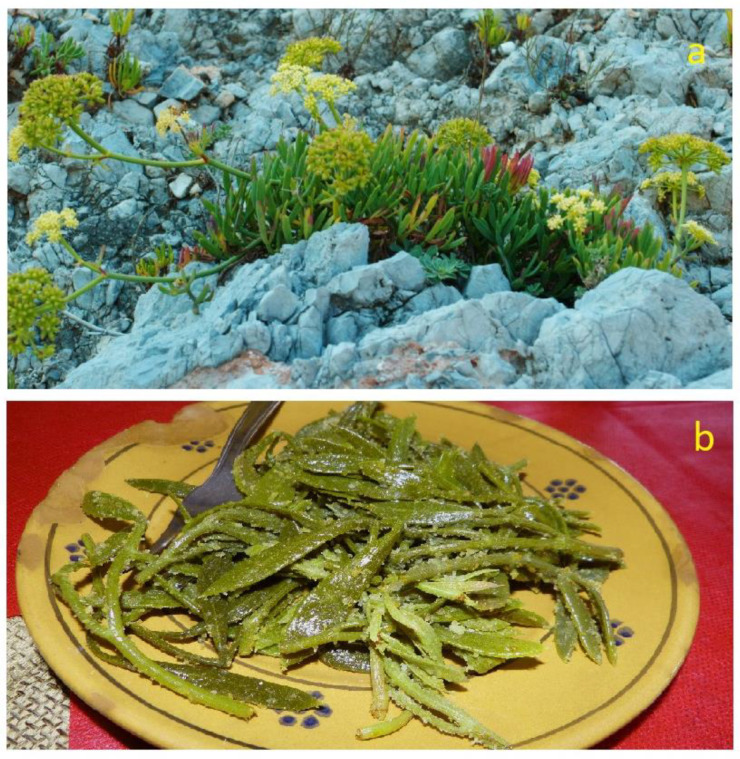
*Crithmum maritimum* (**a**); a typical side dish with sea fennel (**b**), consumed in Salento (Southern Apulia).

**Figure 11 plants-12-00549-f011:**
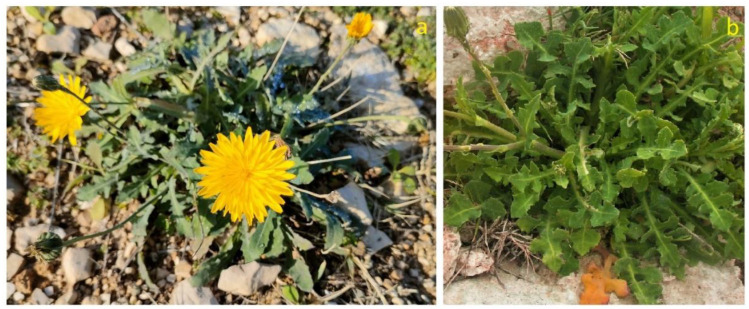
*Reichardia picroides*: flower heads (**a**) and basal rosette (**b**).

**Figure 12 plants-12-00549-f012:**
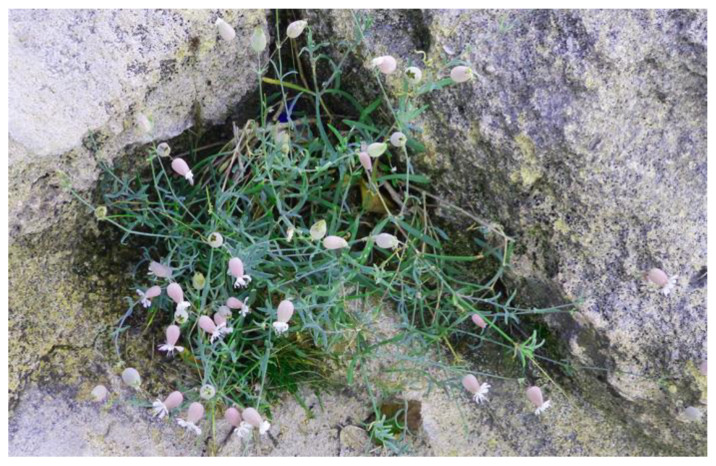
*Silene vulgaris* subsp. *tenoreana*.

**Figure 13 plants-12-00549-f013:**
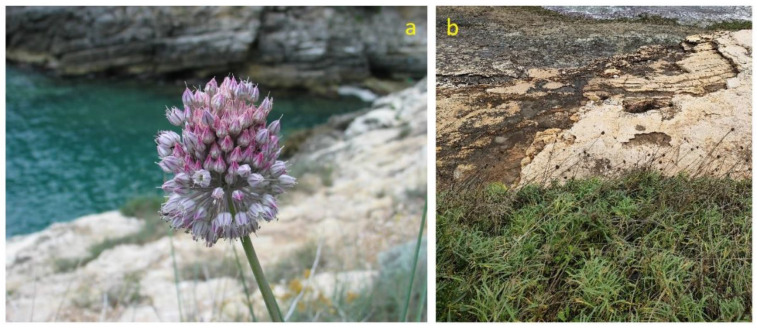
*Allium commutatum* inflorescence (**a**); *A. commutatum* community in the winter season (**b**).

**Figure 14 plants-12-00549-f014:**
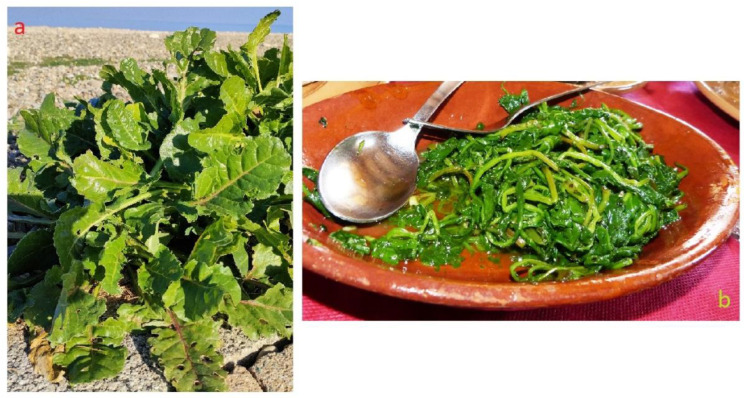
*Beta vulgaris* subsp. *maritima* (**a**); a typical side dish with sea beet (**b**).

**Figure 15 plants-12-00549-f015:**
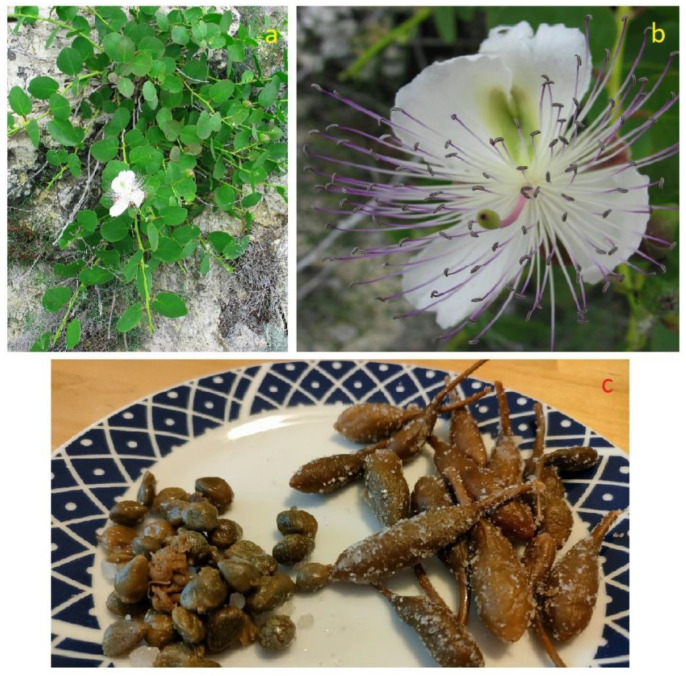
*Capparis spinosa* (**a**) and a detail of the flower (**b**); caper buds and fruits in salt (**c**).

**Table 1 plants-12-00549-t001:** Summary of the main features of the taxa considered in this contribution.

Taxon	Life Form	Geoelement	Protection	Use	Propagation
*Salicornia patula*	T scap	W-European	regional level	food/medical	generative
*Salicornia emerici*	T scap	Steno-Medit.	no	food/medical	generative
*Salicornia dolichostachya*	T scap	Steno-Medit.	no	food/medical	generative
*Salicornia perennis*	Ch succ	Euri-Medit.	regional level	food/medical	generative
*Salicornia fruticosa*	Ch succ	Euri-Medit.	no	food/medical	generative
*Arthrocaulon macrostachyum*	Ch succ/P succ	Medit.	no	food/medical	generative/vegetative
*Soda inermis*	T scap	Paleotemp.	no	food/medical/industrial	generative
*Cakile maritima*	T scap	Medit.-Atl. (Steno-)	no	food/medical	generative
*Crithmum maritimum*	Ch suffr	Euri-Medit./Steno-Medit.	regional level	food/medical	generative
*Reichardia picroides*	H scap	Steno-Medit.	no	food/medical	generative
*Silene vulgaris* subsp. *tenoreana*	H scap	Paleotemp./Subcosmop.	no	food/medical/phytoremediation	generative
*Allium commutatum*	G bulb	Steno-E-Medit.	LC (Least Concern)	food/medical	generative/vegetative
*Beta vulgaris* subsp. *maritima*	H scap	Euri-Medit.	no	food/medical	generative
*Capparis spinosa*	NP	Eurasiat.	regional level	food/medical	generative/vegetative

## Data Availability

Not applicable.

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
