# Peer review of "Edible Halophytes and Halo-Tolerant Species in Apulia Region (Southeastern Italy): Biogeography, Traditional Food Use and Potential Sustainable Crops"

_plants, 2023, doi:10.3390/plants12030549_

Round 1

Reviewer 1 Report

I have read with great pleasure the manuscript which deals with a subject of interest to readers from various fields. The work combines in a  successful way botany, biogeography and ethnobotany.

I would like to suggest only a few minor changes that could improve some aspects of the presentation.

Firstly, the manuscript appears as a review when the structure is that of an article. I suggest that the category be changed from review to article. If it is kept as a review, the structure in Introduction, Results and Materials and Methods should be removed.

The figures could be improved, in Figure 1 next to the general map of Italy a physical or vegetation map of the region studied could be included.

Figures consisting of a single photograph are not very attractive, better to combine several related images in panels with different lettering.

Better not use italics for scientific names of the families, but the generic names should be always in italics and in some cases, they are not written as such (e.g. in line 154, 36). The same for associations (428-430).

For taxonomy I suggest the use of the APG IV system (Amaranthaceae instead of Chenopodiaceae).

Author Response

We would like to thank the Reviewer 1 for the thoughtful comments and efforts towards improving our manuscript. In the following, the Reviewer specific comments and our effort to address these concerns.

 Rev 1

  1. Firstly, the manuscript appears as a review when the structure is that of an article. I suggest that the category be changed from review to article. If it is kept as a review, the structure in Introduction, Results and Materials and Methods should be removed.

                R: according to the reviewer’s suggestion, we changed the category from review to article.

  1. The figures could be improved, in Figure 1 next to the general map of Italy a physical or vegetation map of the region studied could be included.

                R: we have changed the Figure 1 accordingly, by adding a physical map of Apulia next to the general map of Italy

  1. Figures consisting of a single photograph are not very attractive, better to combine several related images in panels with different lettering.

                R: according to the reviewer’s observation, we added more photos and combined several related images in panels with different lettering

  1. Better not use italics for scientific names of the families, but the generic names should be always in italics and in some cases, they are not written as such (e.g. in line 154, 36). The same for associations (428-430).

                R: we made a check of the text and changed accordingly.

  1. For taxonomy I suggest the use of the APG IV system (Amaranthaceae instead of Chenopodiaceae).

                R: we changed the name of the Chenopodiaceae family accordingly.

Reviewer 2 Report

Please, refer to the attached file

Author Response

We would like to thank the Reviewer 2 for the thoughtful comments and efforts towards improving our manuscript. In the following, the Reviewer specific comments and our effort to address these concerns.

Rev 2

  1. All the collected information could be summarized in a table, as suggested in the example below:

R: we added a synthetic table just before the Conclusion section (Table 1), summarizing the main features of the threated taxa.

  1. Lines 181-182 "The most common elements found are Na, Ca, K and Mg, among others, and are present in stem, leaves and roots". The use of the term leaves could be inappropriate given that the leaves are highly reduced and imbricated in these plants.

R: according to the Reviewer’s observation, we changed the sentence (and avoided the term “leaves”)

  1. The titles of the subchapters 3.3.1 and 3.4.1 need to be included.

R: we added the titles

  1. Please consider formatting horizontally the pages containing the three large tables in appendix A.

R: according to the Reviewer’s observation, we formatted horizontally the pages containing the three large tables in appendix A

  1. The manuscript would beneficiate from more figures and morphological illustrations

R: according to the reviewer’s observation, we added more photos, also of morphological plant features.

  1. It would be interesting to receive more information on the investigations carried out in the studied area. Authors might consider showing the questionnaire structure in the supplementary materials. Furthermore, it would be helpful to know when and how many questionnaires were administered and how the authors assembled and analysed the collected data.

R: we added in Appendix B the questionnaire structure and added, in M & M, more information about the data collection

  1. Put the names of the investigated species in the abstract or, at least, in the keywords. Abstracts and keywords are the first elements researchers read when doing bibliographic research. If the authors want their article to reach a broader range of readers, please make the abstract and keywords more informative.

R: according to the reviewer’s observation, we added the names of the investigated species in the abstract and in the keywords

  1. Lastly, the abstract and the main text contain several typos and syntactic errors; therefore, the authors are encouraged to reread the manuscript to correct them. Correct the few scientific names that are not written in Italic.

R: we made a check throughout the manuscript, and corrected numerous typos and syntactic errors; the scientific names (except family names) are now all in italics.
